# Zero-Shot Transfer of Neural ODEs

**Tyler Ingebrand, Adam J. Thorpe, Ufuk Topcu**
University of Texas at Austin
Austin, TX 78712

## Abstract

Autonomous systems often encounter environments and scenarios beyond the scope of their training data, which underscores a critical challenge: the need to generalize and adapt to unseen scenarios in real time. This challenge necessitates new mathematical and algorithmic tools that enable adaptation and zero-shot transfer. To this end, we leverage the theory of function encoders, which enables zero-shot transfer by combining the flexibility of neural networks with the mathematical principles of Hilbert spaces. Using this theory, we first present a method for learning a space of dynamics spanned by a set of neural ODE basis functions. After training, the proposed approach can rapidly identify dynamics in the learned space using an efficient inner product calculation. Critically, this calculation requires no gradient calculations or retraining during the online phase. This method enables zero-shot transfer for autonomous systems at runtime and opens the door for a new class of adaptable control algorithms. We demonstrate state-of-the-art system modeling accuracy for two MuJoCo robot environments and show that the learned models can be used for more efficient MPC control of a quadrotor.

## 1 Introduction

Models that are adaptable, generalizable, and capable of learning online from minimal data are essential for autonomy. These models must adapt to unseen tasks and environments at runtime without relying upon a priori parameterizations. For example, consider an autonomous UAV delivery robot navigating in a dense, urban environment through varying wind patterns and carrying uncertain payloads. This scenario requires rapid adaptation to ensure safe and correct operation because conditions can change unpredictably and online model updates are impractical. While prior works can control autonomous systems in a single setting, they fail to adapt to the continuum of real-life scenarios. The key challenge is enabling *zero-shot transfer* of learned models, where models quickly adapt to new data provided at runtime without retraining.

We present a method for modeling differential equations by learning a set of basis functions parameterized by neural ODEs. Our key insight is to learn a *space* of functions that captures feasible behaviors of the system. By focusing on learning the structure of the space of differential equations, our approach implicitly learns how the dynamics change due to changes in the environment. Our approach is based on the theory of function encoders [14], a framework for zero-shot transfer that has been applied to task transfer in reinforcement learning contexts.

We formulate the space of learned functions as a linear space equipped with an inner product (e.g. a Hilbert space), and learn a set of basis functions over this space, where each basis function is represented by a neural ODE. This structure offers an efficient way to approximate online dynamics via a linear combination of the basis functions.

By representing functions in a Hilbert space and pre-training on a suite of functions, we can quickly identify the basis functions' coefficients for a new dynamical system at runtime using minimal data. This is useful, for instance, in scenarios where we can pre-train offline in simulation, but need to

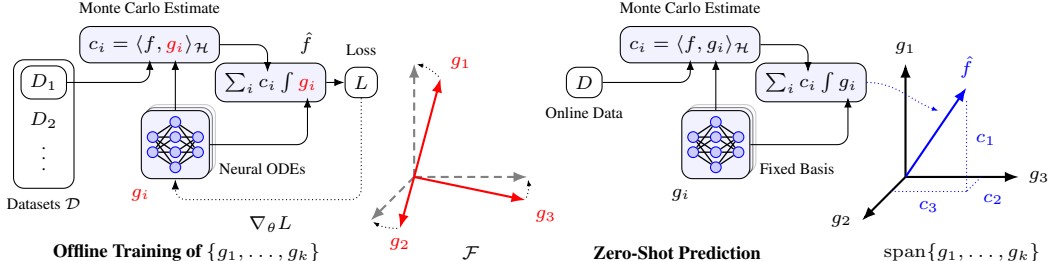

Figure 1: An illustration of our approach. The training phase uses a set of datasets $\mathcal{D}$ to train basis functions $\{g_1, ..., g_k\}$ to span $\mathcal{F}$. The zero-shot phase uses online data to identify the coefficients for a new function, which can be estimated as a linear combination of the basis functions.

quickly identify dynamics online from a single trajectory in a zero-shot manner. This strategy greatly reduces the computational overhead associated with adapting or re-training a neural ODE to new tasks at runtime. The ability to efficiently encode the behavior of a system at runtime without retraining is a key component of our approach. Our approach is outlined in Figure 1.

Our approach yields models which generalize to a large set of possible system behaviors and achieves better long-term prediction accuracy than neural ODEs alone. We showcase our approach on a problem of predicting the behavior of a first-order ODE system with Van der Pol dynamics. We demonstrate the scalability of our approach on two MuJoCo robotics environments [28]. Finally, we test the feasibility of using the learned model for downstream tasks such as model-predictive control (MPC). Our results show that our model achieves significantly better long-horizon prediction accuracy compared to the nearest baseline. Additionally, the MPC controller using our model has a lower slew rate, indicating that the improved model accuracy leads to more efficient control decisions.

## 1.1 Contributions

**Representing Spaces of Dynamical Systems:** We propose a novel framework for representing spaces of dynamical systems, i.e. induced by hidden system parameters, variations in the underlying physics, or changing environmental features. Using a large-scale set of data collected offline, we learn a collection of neural networks that act as a functional basis. This approach is based in the theory of function encoders [14]. Yet the extension to using neural ODEs as basis functions is non-trivial and poses several challenges, mainly from the need to integrate the learned model.

**A Method for Online Adaptation:** We construct a method for adapting neural ODE estimations based on online data without gradient updates, i.e. zero-shot transfer of system models. Our approach overcomes a significant challenge in learning behaviors of differential equations and dynamical systems. By offloading the computational effort to the training phase, we enable rapid online identification, adaptation, and prediction without retraining.

**Empirical Results:** We demonstrate accurate long-horizon predictions in challenging robotics tasks and show these models can be used for online control of quadrotor systems. We assess the quality of the approach in three areas and answer the following questions: 1) How well does our approach adapt to new dynamics online? 2) How does our approach compare to existing approaches for long-horizon prediction tasks? and 3) Does our approach work for downstream tasks such as control? We show that function encoders using neural ODEs as basis functions consistently outperform existing approaches.

## 2 Background

### 2.1 Neural ODEs

Consider an ordinary differential equation $\dot{x}(t) = f(x(t), t)$, where $f$ is Lipschitz continuous and $x(t) \in \mathbb{R}^n$ is the state at time $t$. Given an initial condition $x(t_0)$, the ODE solution can be written as

$$x(t_f) = x(t_0) + \int_{t_0}^{t_f} f\big(x(\tau), \tau\big) d\tau. \tag{1}$$

Note that in general, the explicit dependence of $f(x(t), t)$ on $t$ can be removed by augmenting the state $x$ to include $t$. As such, we omit $t$ throughout.

Neural ODEs [5] parameterize the function $f$ as a neural network. In particular, neural ODEs solve the ODE using an off-the-shelf integrator and optimize the neural network with respect to a prediction loss. The training procedure requires a dataset $D = \{(t_i, x(t_i))\}_{i=1}^d$ which is used to train the model via a supervised objective, such as mean squared error, back-propagated through the integrator.

Neural ODEs have demonstrated impressive accuracy for long-horizon predictions of continuous-time systems. Furthermore, they can be trained on trajectories with irregular time intervals between samples [18], and generalize better than multi-layer perceptron models. However, they lack adaptability and need to be retrained for every scenario.

## 2.2 Function Encoders

To achieve zero-shot transfer, we employ the theory of *function encoders* [14]. While typical machine learning approaches learn a single function, function encoders learn basis functions to span a *space* of functions. This allows the function encoder to achieve zero-shot transfer within this space by identifying the coefficients of the basis functions for any function in the space at runtime. Once the coefficients have been identified, the function can be reproduced as a linear combination of basis functions. Despite the broad applicability of function encoders, the extension to capture solutions to differential equations via neural ODEs is non-trivial.

Formally, consider a function space $\mathcal{F} = \{f \mid f : \mathcal{X} \to \mathbb{R}^m\}$ where $\mathcal{X} \subset \mathbb{R}^n$. Instead of learning a single function, function encoders learn $k$ basis functions $g_1, g_2, \ldots, g_k$ that are parameterized by neural networks in order to span $\mathcal{F}$ [14]. Define the inner product of $\mathcal{F}$ as $\langle f, g \rangle_{\mathcal{F}} = \int \langle f(x), g(x) \rangle dx$. Then, the functions $f \in \mathcal{F}$ can be represented as a linear combination of basis functions,

$$f(x) = \sum_{i=1}^k c_i g_i(x \mid \theta_i), \tag{2}$$

where $c \in \mathbb{R}^k$ are real coefficients and $\theta_i$ are the network parameters for $g_i$. Let $V$ be the volume of $\mathcal{X}$. For any function $f \in \mathcal{F}$, an empirical estimate of the coefficients $c$ can be calculated using data $\{(x_j, f(x_j))\}_{j=1}^m$ via a Monte-Carlo estimate of the inner product,

$$c_i = \langle f, g_i \rangle \approx \frac{V}{m} \sum_{j=1}^m \langle f(x_j), g_i(x_j \mid \theta_i) \rangle. \tag{3}$$

The basis functions are trained using a set of datasets, $\mathcal{D} = \{D_1, D_2, \ldots\}$, where each dataset $D_i = \{(x_j, f_i(x_j)\}_{j=1}^m$ consists of input-output pairs corresponding to a single function $f_i \in \mathcal{F}$. For each function $f_i$ and corresponding dataset $D_i$, first compute the coefficients $\{c_1, c_2, \ldots, c_k\}$ via (3) and then obtain an empirical estimate of $f_i$ via (2). Then compute the error of the estimate of $f_i$ though the norm induced by the inner product of $\mathcal{F}$. The loss function is simply the sum of the losses for all $f_i$, which is minimized via gradient descent. For more details, see [14].

After training, the basis functions are fixed, and the coefficients $c$ of a new function $f \in \text{span}\{g_1, \ldots, g_k\}$ are computed via (3) or via least-squares, using data collected online. This is key for efficient, online calculations, since the approximation in (3) is effectively a sample mean and requires no gradient calculations.

**Orthogonality of the Basis Functions:** Note that function encoders do not enforce orthogonality of the basis functions explicitly [14]. Using Gram-Schmidt to orthonormalize the basis functions during training can significantly increase the training time and is computationally intensive. Instead, during training the coefficients are computed using (3) presuming that the basis functions are orthogonal. This is key. This causes the basis vectors to naturally become more orthogonal as training progresses since the loss implicitly penalizes the basis functions if they are not orthonormal. See Appendix I.

## 3 Function Encoders With Neural ODEs as Basis Functions

Consider a space $\mathcal{F}$ of Lipschitz continuous dynamical systems $f : \mathcal{X} \to \mathcal{X}$. The space of dynamical systems can arise, for instance, due to uncertain parameters, minor variations in a first-order physics

model, or changing environmental features. Given an initial condition $x(t_0)$, our goal is to estimate the state $x(t_f)$ at a future time $t_f > t_0$. The integral form of the initial value problem is given by (1).

Our approach can be separated into two distinct phases: *offline training* and *zero-shot prediction*. During *offline training*, we presume that we have access to an offline dataset $\mathcal{D} = \{D_1, D_2, \ldots\}$, where $D_i$ is a realization of a trajectory from a function $f_i \in \mathcal{F}$. During *zero-shot prediction*, we seek to predict a previously unseen function $f$ and have access to a minimal trajectory $D$ taken from $f$. We seek to learn a set of basis functions $g_1, \ldots, g_k$ that span $\mathcal{F}$, where $k$ is a user-specified hyper-parameter. By learning a set of basis functions that span the space $\mathcal{F}$, we obtain a means to represent the behavior of any dynamical system in the space. However, we do not observe measurements of $f$ directly since we cannot typically measure the instantaneous derivative $\dot{x}$ of a dynamical system. Instead, we will equivalently learn a set of neural ODE basis functions such that the underlying neural networks which are being integrated correspond to $g_1, ..., g_k$. We then seek to compute a representation of a new function using data collected online.

### 3.1 Computing a Set of Neural ODE Basis Functions

For every $f \in \mathcal{F}$, we define the integral term in (1) as a function $H : \mathcal{X} \times \mathcal{T} \to \mathcal{X}$, given by,

$$H\big(x(t_0), t_f\big) := \int_{t_0}^{t_f} f\big(x(\tau)\big) d\tau. \tag{4}$$

We model the dynamical system $f$ using a function encoder as in (2). Using (2) in (4), and by the linearity of the definite integral, we have that,

$$H\big(x(t_0), t_f\big) = \int_{t_0}^{t_f} \left[ \sum_{i=1}^{k} c_i g_i\big(x(\tau) \mid \theta_i\big) \right] d\tau = \sum_{i=1}^{k} c_i \int_{t_0}^{t_f} g_i\big(x(\tau) \mid \theta_i\big) d\tau = \sum_{i=1}^{k} c_i G_i\big(x(t_0), t_f\big), \tag{5}$$

where $g_i$ is neural network parameterized by $\theta_i$ and $G_i(x(t_0), t_f) := \int_{t_0}^{t_f} g_i(x(\tau) \mid \theta_i) d\tau$. One interpretation of the above equation is that we can represent $H$ as a weighted combination of basis functions $G_i$, and the problem of learning a set of basis functions $g_1, \ldots, g_k$ can equivalently be viewed as learning a set of neural ODEs $G_1, \ldots, G_k$. Thus, we define the Hilbert space $\mathcal{H}$ of functions $H$ as in (4) and equip it with the following inner product,

$$\langle H, G \rangle_{\mathcal{H}} := \int \langle H(z, t), G(z, t) \rangle_{\mathcal{X}} d(z, t). \tag{6}$$

We then learn basis functions $G_1, \ldots, G_k$ spanning $\mathcal{H}$ where each basis function is a neural ODE.

From (6), the coefficients of a function $H \in \mathcal{H}$ are given by $c_i = \langle H, G_i \rangle_{\mathcal{H}}$. However, from [14], computing the inner product exactly is generally intractable in high-dimensional spaces. We can empirically estimate the coefficients $c_i$ using a trajectory of (potentially irregularly) sampled states $\{x(t) \mid t = t_0, \ldots, t_m\}$ from a dynamics function $f \in \mathcal{F}$. Using the trajectory, we form the dataset $D = \{(x(t_j), x(t_{j+1})\}_{j=0}^{m-1}$. We can compute $c$ using $D$ via a Monte-Carlo estimate of the inner product in (6),

$$c_i = \langle H, G_i \rangle_{\mathcal{H}} \approx \frac{V}{m} \sum_{j=0}^{m-1} \left\langle x(t_{j+1}) - x(t_j), G_i\big(x(t_j), t_{j+1}\big) \right\rangle_{\mathcal{X}}, \tag{7}$$

where $V$ is the volume of the region of integration, and following from (1),

$$x(t_{j+1}) - x(t_j) = \int_{t_j}^{t_{j+1}} f(\tau) d\tau = H\big(x(t_j), t_{j+1}\big). \tag{8}$$

In other words, we substitute the difference between states $x(t_{j+1}) - x(t_j)$ for $H(x(t_j), t_{j+1})$ in (7).

Let $\mathcal{D} = \{D_1, D_2, ...\}$ be a set of datasets, where each $D_\ell = \{(x(t_j), x(t_{j+1}))\}_{j=0}^{m-1}$ is collected from a trajectory from a function $f_\ell \in \mathcal{F}$. For each dataset $D_\ell$, we compute the coefficients $c_1, \ldots, c_k$ according to (7). The coefficients can be used to approximate the corresponding $H_\ell$ via (5). We then evaluate the error of $H_\ell$ using the dataset $D_\ell$ and minimize its loss via gradient descent. This is done

**Algorithm 1** Training Function Encoders with Neural ODE Basis Functions

---

1: **Input:** Set of datasets $\mathcal{D}$, number of basis functions $k$, learning rate $\alpha$
2: **Output:** Neural ODE basis functions $G_1, G_2, ..., G_k$
3: Initialize $g_1, g_2, ..., g_k$ as neural networks with parameters $\theta = \{\theta_1, \theta_2, ..., \theta_k\}$
4: **while** not converged **do**
5:     loss $L = 0$
6:     **for all** $D_\ell \in \mathcal{D}$ **do**
7:         **for** $i \in 1, ..., k$ **do**
8:             $c_i \approx \frac{V}{m} \sum_{j=0}^{m-1} \langle x(t_{j+1}) - x(t_j), G_i(x(t_j), t_{j+1} - t_j) \rangle_{\mathcal{X}}$
9:         **end for**
10:        $L = L + \sum_{j=0}^{m-1} \|(x(t_{j+1}) - x(t_j)) - \sum_{i=1}^{k} c_i G_i(x(t_j), t_{j+1} - t_j)\|^2$
11:     **end for**
12:     $\theta = \theta - \alpha \nabla_\theta L$
13: **end while**

---

for multiple functions $f \in \mathcal{F}$ at each gradient update to ensure the basis learns the space rather than a single function. We present this as Algorithm 1.

Applying Algorithm 1 yields basis functions which span the space of dynamical systems, where each basis function is a neural ODE. This space describes possible behaviors of the system, where variations in environmental parameters, physics, etc. correspond to a particular dynamics function within this space. Therefore, this algorithm represents complicated system behaviors simply as a vector within a Hilbert space. Section 3.2 shows how to use these basis functions for zero-shot dynamics prediction from small amounts of online data.

## 3.2 Efficient Online Transfer Without Retraining

After training, we fix the parameters of the basis functions $g_1, ..., g_k$, and can compute the coefficient representation $c \in \mathbb{R}^k$ for any function $f \in \text{span}\{g_1, ..., g_k\}$ via (7). If $\mathcal{D}$ is rich enough to capture the various behaviors of the systems in $\mathcal{F}$, then we can estimate the behavior of any dynamics $f \in \mathcal{F}$.

Given data collected online from a single trajectory, we can compute the coefficients using the Monte-Carlo estimate of the inner product as in (7). This approximation is a crucial component of the approach. It allows the inner product to be computed from data through an operation that is effectively a sample mean. Therefore, this approach can be computed online quickly even for large amounts of data. Then, given the coefficients, the future states of the system can be predicted using (5). These properties allow the neural ODE to achieve zero-shot transfer. Identifying the coefficients only requires inner product calculations, vector addition, and scalar multiplication, and so it can be computed online without any gradient updates.

**The Residuals Method:** The zero vector of the coefficients space corresponds to the zero function. Since the feasible dynamics are differentiated by their coefficients, it is numerically convenient if the coefficients corresponding to all feasible systems are centered around zero.

Thus, we can re-center the space of coefficients around the center of the *cluster* of feasible dynamics. This is done by first modeling the average dynamics $F_{avg}$ in the set of datasets $\mathcal{D}$, and then learning the residuals between each function and $F_{avg}$. In other words, the basis functions are trained to span the function space corresponding to $R(x(t_0), t_f) = x(t_f) - x(t_0) - F_{avg}(x(t_0), t_f)$. This method can achieve better accuracy, but requires learning one additional neural ODE, $F_{avg}$. Alternatively, an approximate dynamics model based on prior knowledge can be used as $F_{avg}$. We describe the training procedure for this approach in Algorithm 2.

## 3.3 Incorporating Zero-Order Hold Control Inputs

We can account for a zero-order hold (ZOH) control input $u \in \mathcal{U} \subset \mathbb{R}^p$ with minimal modifications. A ZOH control input is given by a piecewise constant function, meaning it is held constant over the period of integration. Given controlled dynamics, $f : \mathcal{X} \times \mathcal{U} \to \mathcal{X}$, we modify the corresponding functions $H$ to incorporate a constant input, $H(x(t_0), u, t_f) = \sum_{i=1}^{k} c_i \int_{t_0}^{t_f} g_i(x(t_0), u \mid \theta_i) d\tau$. Then, using trajectory data that also includes the controls applied at each time interval, we can estimate

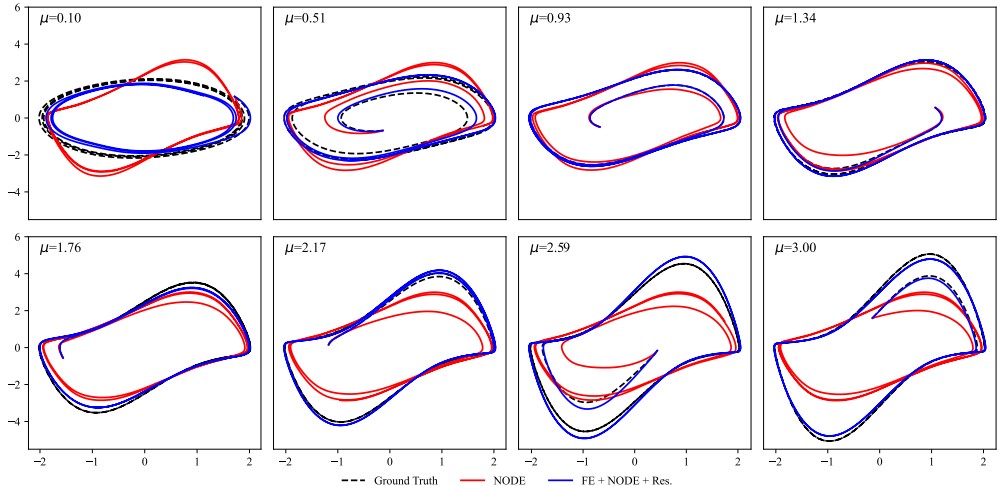

Figure 2: The approximated dynamics for different Van der Pol systems, where the parameter $\mu$ is varied. This plot shows that a NODE can only fit a single Van der Pol system, whereas `FE + NODE + Res` can fit a space of Van der Pol systems from $5000$ example data points.

the coefficients using datasets $D = \{(x(t_j), u_j, x(t_{j+1}))\}_{j=0}^{m-1}$, substituting $g_i\big(x(\tau), u, \,|\, \theta_i\big)$ in (5) and (7). The remainder of the training procedure is unchanged.

## 4 Numerical Experiments

We demonstrate the effectiveness of our approach for predicting and controlling dynamical systems through several numerical experiments. We first demonstrate that the approach can adapt to different dynamics using a Van Der Pol oscillator system. We then show long-horizon prediction accuracy on challenging MuJoCo robotics experiments and compare to neural ODEs (`NODE`) [5] and function encoders as in [14] using the residuals method (`FE + Res`). Lastly, we show the learned models are sufficiently accurate for downstream tasks on a difficult control task using a quadrotor system. The source code is available at `https://github.com/tyler-ingebrand/NeuralODEFunctionEncoder`.

Current off-the-shelf integrators with adaptive step sizes do not support efficient batch calculations. Because this algorithm involves training numerous neural ODEs on a large amount of data, the ability to train on data in batches is required. Therefore, we implement an RK4 integrator since it can be efficiently computed for multiple data points in parallel. The Van der Pol visualization uses 11 basis functions while the MuJoCo and Drone experiments use 100. For ablations on how the hyper-parameters affect results, see Appendix G.

### 4.1 Visualization on a Van der Pol Oscillator

We first demonstrate that our approach can adapt to a space of dynamics that vary according to a nonlinear parameter. The Van der Pol dynamics are defined as, $\dot{x} = y$, $\dot{y} = \mu(1 - x^2)y - x$, where $[x, y]^\top \in \mathbb{R}^2$ is the state, and $\mu$ is a hidden parameter. We collect multiple datasets $D_\ell$ where $\mu$ is fixed for the duration of the trajectory, but varies between trajectories. We train basis functions using Algorithm 1. We then compute the coefficients via (7) and approximate the dynamics via (5).

We plot the results in Figure 2. As expected, we observe that our proposed approach can predict the dynamics of a space of Van der Pol systems without retraining. We can also see that a single neural ODE trained on the same data can only fit a single function. Therefore, its prediction corresponds most closely with the behavior of a single Van der Pol system that has the mean $\mu$ value. This illustrates that our approach is capable of adapting to different dynamics at runtime.

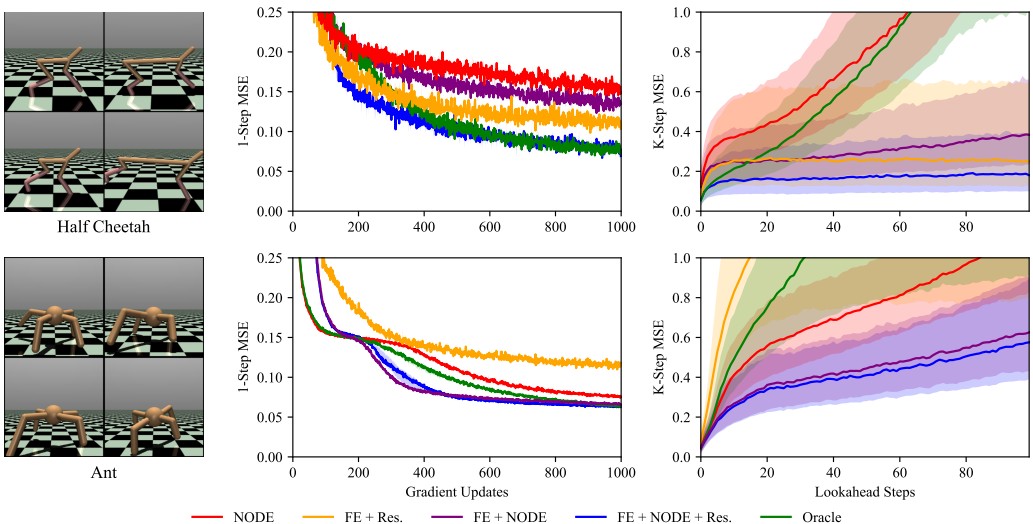

Figure 3: Model performance on predicting the dynamics of MuJoCo robotics environments with hidden parameters. 200 example data points are given to identify dynamics. The results show that `FE + NODE + Res.` makes accurate, long-horizon predictions even in the presence of hidden parameters. Evaluation is over 5 seeds, shaded regions show the first and third quartiles around the median.

## 4.2 Long-Horizon Prediction on MuJoCo Environments

We evaluate the performance of our proposed approach on the Half-Cheetah and Ant environments [28], shown in Figure 3. The hidden environmental parameters are the length of the limbs, the friction coefficient, and the control authority. Of the two environments, Ant is more difficult due to its higher degrees of freedom. For training, we collect a dataset of trajectories where the hidden parameters are unobserved, but held constant throughout the duration of a given trajectory. After training, we use 200 datapoints, equivalent to about seven seconds of data for a system running at 30 Hertz, and use only this data to identify the dynamics. Note this online phase is computationally simple, and can be done in only milliseconds on a GPU.

Neural ODEs (`NODE`) perform poorly because they have no mechanism to condition the prediction on the hidden-parameters. Effectively, `NODE` learns the mean dynamics over all dynamics functions in the training set. Function encoders using the residuals method (`FE + Res`) can implicitly condition their predictions on the hidden parameters through the coefficient calculation, though they are unable to achieve accurate long horizon predictions on the more challenging Ant problem. This is because it lacks the inductive bias of neural ODEs. Our approach (`FE + NODE`) can both implicitly condition the predictions on the hidden parameters through data, but also benefits from the inductive bias of neural ODEs. We see that the residuals method performs best out of all approaches in both environments. This is because the average model significantly reduces the epistemic uncertainty and provides a meaningful baseline from which to center the training. The average model acts as a good inductive bias and makes it easier to distinguish between the learned functions during training. We additionally compare against an oracle prediction approach (`Oracle`), which has access to the hidden parameters as an additional input with a neural ODE as the underlying architecture. While its 1-step prediction accuracy demonstrates good empirical performance, the long-horizon predictions are unstable. This is because `Oracle` is required to generalize to an entire space of dynamics with one NODE, which is a complex and difficult function to learn.

## 4.3 Realistic Robotics Experiments and Control of a Quadrotor System

Lastly, we seek to test the accuracy of our approach for use on downstream tasks such as control on a realistic example using a robotic system. We seek to determine if the learned models are sufficiently accurate for model-based control in the presence of hidden parameters. We use a simulated quadrotor system using PyBullet [31], which is a highly nonlinear control system. We use the quadrotor's mass as a hidden parameter. The goal is to predict the future state of the quadrotor system under any hidden

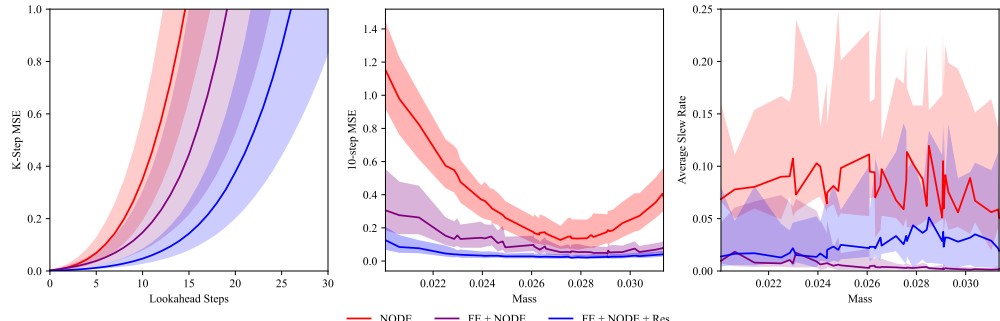

Figure 4: Model performance on the PyBullet quadrotor environment with varying mass. Function encoders improve model performance across varying masses. Shaded region is $1^{st}$ and $3^{rd}$ quartiles over 200 trajectories (left) and over 5 trajectories (middle, right).

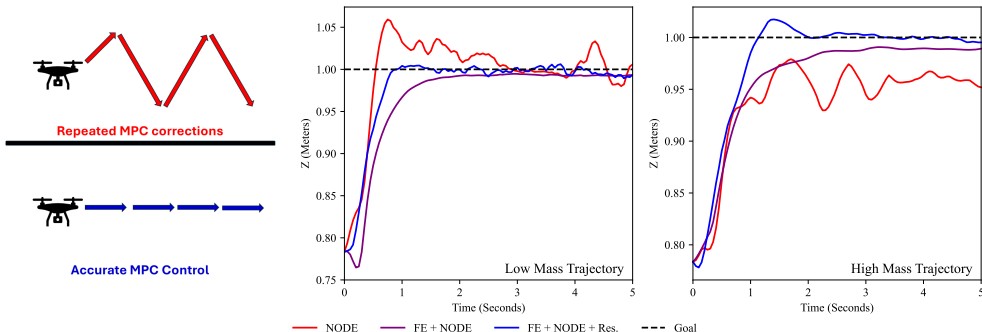

Figure 5: Qualitative analysis of the difference in control between NODEs and our approach. Two trajectories with the same initial position but different masses are shown. `NODE` is unaware of the mass, and so its $z$ position requires constant correction. In contrast, `FE + NODE (+Res)` accounts for the mass through the coefficients, meaning it is more accurate and requires fewer corrections.

parameters, using 2000 data points collected online to identify the dynamics. Then, given the learned model of system behavior, we seek to control the quadrotor using gradient-based model predictive control (MPC) to reach a pre-specified hover point. The results are plotted in Figure 4.

The results show that `FE + NODE + Res` outperforms competing approaches at long-horizon predictions. Furthermore, we plot the 10-step MSE as a function of mass in Figure 4, and we observe that `FE + NODE + Res` accurately predicts the system behavior across varying masses. We observe a slight decay in performance for low masses which are more sensitive to control inputs during simulation, which causes the simulated trajectories to diverge more from the rest of the observed data. The neural ODE (`NODE`) performs poorly for different masses, and its performance decays quickly as the dynamics deviate from the mean behavior.

Lastly, we see that this prediction accuracy translates to the downstream performance of an MPC controller. While all approaches are sufficiently accurate for control due to the fact that MPC is partially robust to model inaccuracies, the prediction accuracy has a corresponding impact on the task performance. Neural ODEs (`NODE`) demonstrate a high slew rate, which reflects the need for repeated positional corrections necessitated by taking bad actions. In contrast, `FE + NODE` and `FE + NODE + Res` have lower slew rates as they make more accurate decisions. We plot two example trajectories that demonstrate this behavior in Figure 5.

## 5   Scope & Limitations

**Overhead:** Our approach incurs a cost of either increased inference time or memory, depending on if the basis functions are integrated sequentially or in parallel. We integrate them sequentially during

training to reduce memory overhead and allow for larger batch sizes, while integrating in parallel at execution to prioritize inference speed. See Appendix D.

**Data dependency:** In order to make efficient, online approximations of a new function $f \in \mathcal{F}$ without gradient calculations, the basis functions must be trained to span the space of possible dynamics. To do so, there must be sufficient example datasets of possible dynamics under fixed hidden parameters in $\mathcal{D}$. This implies a larger amount of data must be collected to learn a space of dynamics then would be needed to learn a single dynamics function.

**Integration:** The training procedure trains the basis functions on short time intervals $t_f - t_0$, in which $x(t) \in \mathcal{X}$. The basis functions have only been trained for inputs in the space of $\mathcal{X}$, where their behavior outside of $\mathcal{X}$ is unpredictable. As a result, it is necessary to either integrate each basis function for a short time interval before calculating the state according to (5), or to integrate the basis functions as described in B. Integrating the basis functions over long horizons without calculating the state of the system during intermediate steps may lead the predicted state to leave $\mathcal{X}$, at which point the behavior of that basis function becomes unpredictable.

## 6 Related Work

**Basis Functions:** Function approximation techniques often employ a linear model over a predefined set of basis functions. Techniques such as Taylor series, Fourier series, and orthogonal polynomial systems utilize an infinite set of basis functions, theoretically allowing perfect function representation [6, 23, 7]. However, in high-dimensional spaces, these techniques become impractical due to the exponential growth in the number of basis functions. Additionally, many approaches depend on the choice of a feature map or kernel to define the function space [29, 25], which imposes structure by selecting the class of functions to learn from (e.g. using radial basis functions [3]). These design choices necessarily introduce approximation errors through the choice of function class, or may depend on prior domain knowledge, which may not always be available. Methods for identifying system dynamics that use a large library of pre-defined basis functions, such as Koopman operators [2] or nonlinear system identification through sparse regression such as SINDy [4, 26, 15], have received considerable attention. Yet these approaches typically employ a finely-crafted finite dictionary of basis functions, which requires careful choice to achieve good data-driven performance [20]. Neural network approaches such as functional-link and orthogonal networks [30, 7] omit hidden layers and use gradient descent to learn a linear combination of features, encoding the function class into the network architecture, but fail to generalize well, and are not amenable to zero-shot transfer. In contrast, we compute the coefficients of the model through a well-defined inner product, which scales well with data and can be computed quickly. Furthermore, our basis functions are entirely learned from data during the training phase, similar to representation learning [1], and thus require no prior assumptions or domain knowledge.

**Neural ODEs:** In existing work, neural ODEs have proven to be a powerful tool for modeling dynamical systems [10, 24, 22, 11, 17] and stochastic differential equations [21, 13], but generally require an extensive data collection and training phase. While the model training can be enhanced [8] and the models can incorporate prior knowledge [9, 10] to reduce the training time, they inherently focus on a single system at a time. This inherently limits their ability to generalize across different systems without retraining. Notably, parameterized neural ODEs [19] pass the model parameters as an additional input to the neural ODE. This approach has been shown to achieve a form of transfer within the set of allowable parameters, but requires extensive knowledge of the system parameters and the structure of the dynamics, both at training and test time. In principle, our approach can be combined with these existing approaches to incorporate their distinct advantages, making our approach highly generalizable to different systems and modeling frameworks.

**Deep Learning Techniques:** Few-shot meta learning aims to solve a similar problem, where a learned model is adapted given an online dataset [12]. However, meta learning requires gradient updates to the learned models, which may be too slow for real-time control. Transformers are another technique that can adapt given an online dataset by feeding that dataset as input to the encoder side of the transformer. However, transformers have long forward pass times and scale quadratically with the amount of data [16], and so they are not amenable for model-based control. Domain randomization in reinforcement learning is another technique to generate a policy which is robust to a large set

of dynamics [27]. In contrast, our dynamics model *adapts* to the current dynamics, and thus our controller is adaptive, rather than robust.

# 7 Conclusion & Future Work

We introduced zero-shot neural ODEs, which accomplish both long-horizon predictions and zero-shot transfer. We demonstrated the performance of this approach on two challenging MuJoCo tasks and on the control of a quadrotor system. Our approach makes a significant step towards online adaptability of model-based control and has implications for the safe control of autonomous systems in the presence of uncertainty. In future work, we plan to address safety during training, perhaps using the properties of the Hilbert space to characterize the epistemic uncertainty. We also plan to explore theoretical extensions to stochastic differential equations and Hilbert spaces of probability measures.

# 8 Broader Impact

This approach demonstrates several clear benefits for enabling same-day adaptation, which is a critical need for autonomous systems that will be deployed in new, unstructured environments. Nevertheless, this approach will require a more thorough theoretical analysis before it can be deployed on actual robotics systems, e.g. to determine confidence or sample bounds to guarantee safety. Notably, this work is a step toward bridging the sim-to-real gap, though it remains unclear how well real-world systems will be represented by a set of basis functions learned in simulation.

# 9 Acknowledgements

Thank you to Dr. Cyrus Neary for helpful discussions. This material is based upon work supported by the National Science Foundation under NSF Grant Number 2214939. Any opinions, findings, and conclusions or recommendations expressed in this material are those of the authors and do not necessarily reflect the views of the National Science Foundation. This material is based upon work supported by the Air Force Office of Scientific Research under award number AFOSR FA9550-19-1-0005. Any opinions, findings and conclusions or recommendations expressed in this material are those of the author(s) and do not necessarily reflect the views of the U.S. Department of Defense.

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

## A  Hardware

All experiments use an Intel 9th Generation i9 CPU and a Nvidia 2060 GPU with 6GB of memory.

## B  Faster Integration

When computing the approximate inner product between the true system and each basis function, it is necessary to compute $G_i(x(t_0), t_f)$ for every tuple in the dataset $D$. However, once the coefficients $c$ have been computed, it is, in theory, no longer necessary to integrate the basis functions separately. From (5), we can approximate $H$ as

$$H(x(t_0), t_f) = \int_{t_0}^{t_f} \left[ \sum_{i=1}^{k} c_i g_i(x(\tau) \mid \phi_i) \right] d\tau, \tag{9}$$

by the linearity of the integral. In effect, we are summing the gradients of each basis function rather than the basis functions themselves. As a result, inference for a specific set of coefficients can be decreased from requiring $k$ integrations to only a single integration. However, we find this method to make less accurate predictions in practice, which may be due to our choice of integrator. This trick may be better suited to variable step-size integrators, which benefit more from reduced calls to the integrator than RK4 does.

## C  Method of Integration

We leverage RK4 as the default integrator for this work, as it can run a forward pass in milliseconds. There are more accurate integrators available, such as odeint. However, there is inherently a trade off with respect to both training time and execution time. Integrators such as adaptive step size solvers can potentially make 20 or more calls to the neural ODE during a forward pass, while RK4 makes only 4. The increased number of neural ODE forward passes greatly increases memory usage and compute time. We experimented with more accurate integrators, but ultimately found this tradeoff to be unfavorable. Future work should investigate integrators that are fast, but achieve better accuracy than RK4.

## D  Overhead

While the function encoder algorithm alone has minimal overhead relative to a MLP, this is not the case for neural ODEs due to the need for integration. The $k$ neural ODEs may either be integrated sequentially or in parallel. If they are integrated sequentially, the memory overhead is lower, especially with respect to back-propagation. Thus, we find this useful for *offline training*, where the sequential method allows us to compute gradients for a larger batch of data at the cost of training time. In contrast, *online execution* generally favors inference speed over memory overhead. Thus, we use the parallel method for online inference in the drone example, which requires much more memory but only a small overhead of inference time relative to neural ODEs alone.

## E  Residuals Method Algorithm

Training a function encoder with the residuals method requires two separate loss functions. The first loss function trains $F_{avg}$ on data from all datasets $D_\ell$, which means it effectively learns the expectation of $F$ given the training set. This loss function can be skipped if $F_{avg}$ is a fixed function based on prior knowledge.

The second loss function trains the basis functions. Unlike in Algorithm 1, the function being learned is $x(t_{j+1}) - x(t_j) - F_{avg}(x(t_j), t_{j+1} - t_j)$. In other words, the residual between the data and the average function. This loss is only used to train the basis functions, it is **not** used to train the average function. See Algorithm 2.

---

**Algorithm 2** The Residuals Method

---

1: **Input:** Set of datasets $\mathcal{D}$, number of basis functions $k$
2: **Output:** Average function $F_{avg}$ and Neural ODE basis functions $G_1, G_2, ..., G_k$
3: Initialize $f_{avg}$ and $g_1, g_2, ..., g_k$ as neural networks with parameters $\bar{\theta}$ and $\theta = \{\theta_1, \theta_2, ..., \theta_k\}$
4: **while** not converged **do**
5:     // Train Average Function
6:     loss $L_1 = 0$
7:     **for all** $D_\ell \in \mathcal{D}$ **do**
8:         $L_1 = L_1 + \sum_{j=1}^{m-1} \|(x(t_{j+1}) - x(t_j)) - F_{avg}(x(t_j), t_{j+1} - t_j)\|^2$
9:     **end for**
10:    $\bar{\theta} = \bar{\theta} - \alpha \nabla_{\bar{\theta}} L_1$
11:    // Train Basis Functions
12:    loss $L_2 = 0$
13:    **for all** $D_\ell \in \mathcal{D}$ **do**
14:        **for** $i \in 1, ..., k$ **do**
15:            $c_i \approx \frac{V}{m-1} \sum_{j=1}^{m-1} \langle x(t_{j+1}) - x(t_j) - F_{avg}(x(t_j), t_{j+1} - t_j), G_i(x(t_j), t_{j+1} - t_j) \rangle_{\mathcal{X}}$
16:        **end for**
17:        $L_2 = L_2 + \sum_{j=1}^{m-1} \| (x(t_{j+1}) - x(t_j) - F_{avg}(x(t_j), t_{j+1} - t_j)) - \sum_{i=1}^{k} c_i G_i(x(t_j), t_{j+1} - t_j) \|^2$

18:    **end for**
19:    $\theta = \theta - \alpha \nabla_\theta L_2$
20: **end while**

---

# F   Implementation Details

All baselines use the same training scheme. We use an ADAM optimizer with a learning rate of $1e-3$, and gradient clipping with a max norm of $1$. NODE baselines uses 4 hidden layers of size 512, while FE + NODE baselines uses 4 hidden layers of size 51 for each basis function. Note this leads to approximately the same number of parameters for both approaches because the number of hidden parameters scales quadratically with the size of the hidden layers. All baselines train on 50 functions per gradient update via gradient accumulation. States are normalized to have 0 mean and unit variance.

A random policy is used to collect data for the MuJoCo environments. A PID-based exploratory policy, which moves to random nearby points, is used to collect data for the quadrotor since a random policy collides with the floor. Evaluations are done on a holdout set collected through the same means.

All quadrotor baselines use the same MPC controller. The controller optimizes the actions through a combined sampling, gradient descent over 100 iterations. The episode is 100 steps, while the planning horizon is 10 steps. The controller optimizes 100 sample trajectories in parallel, and ultimately chooses the best one. Warm starting is used for following MPC calls to improve performance. The cost function penalizes distance to the objective point, deviance from a stable horizontal position, velocity, and the difference between torques on each rotor.

# G Hyper-Parameter Ablations

## G.1 Number of Basis Functions

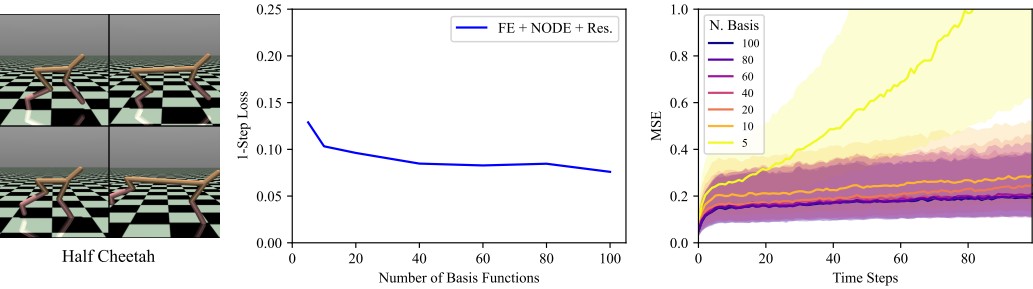

Figure 6: We ablate the effect of the number of basis functions (k) on the performance of the learned model. Results are shown for the FE + NODE + Res. algorithm applied to the Half Cheetah environment. The results indicate the the proposed approach is insensitive to the number of basis functions around $k = 100$, while performance eventually decays as $k$ approaches 0.

## G.2 Number of Example Data Points

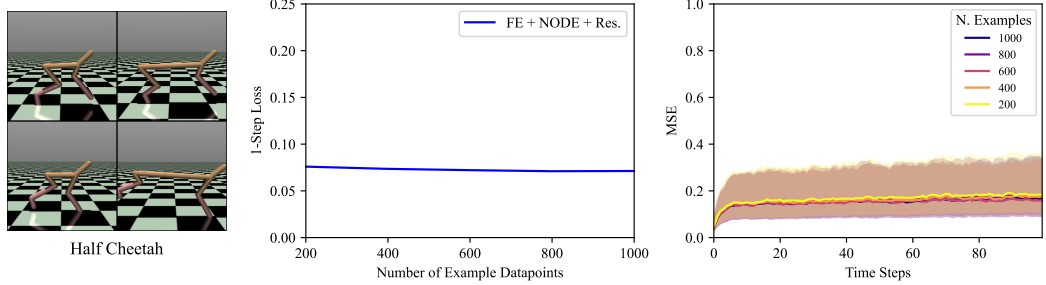

Figure 7: We ablate the effect of the number of example data points on the performance of the learned model. Results are shown for the FE + NODE + Res. algorithm applied to the Half Cheetah environment. The results indicate the the proposed approach is insensitive to increasing example dataset sizes, which suggests that 200 data points is sufficient for the coefficients to converge.

# H Generalization

Figure 3: This figure shows the generalization capabilities of the proposed method. The black line indicates the ground truth Van Der Pol dynamics, and the red line shows an approximation. The model was trained on $\mu \in [0.1, 3.0]$. The left side of the figure shows Van Der Pol dynamics for values of $\mu$ that are within the distribution of training environments, though each environment is unseen. The right shows the approximation for $\mu = 4.0$, which lies outside of the training distribution. The figure shows that the function encoder is able to reasonably generalize outside of its training set in this example.

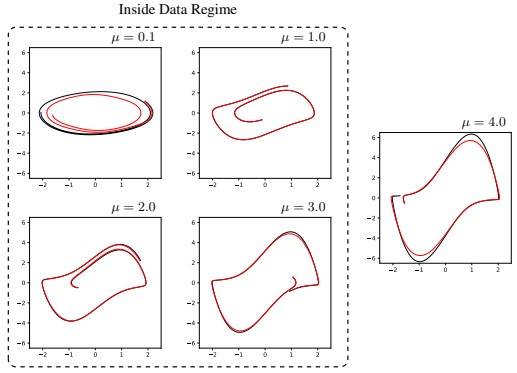

# I  Orthonormality

Consider a set of basis functions $g_1, \ldots, g_k$. Suppose that $g_1, \ldots, g_k$ is not orthonormal. Now consider a function $f$, and suppose $f$ happens to be in the span of $g_1, \ldots, g_k$. Then $f$ can be expressed as $f = b^\top g$, where $b$ is a set of coefficients and $g$ is the concatenation of $g_1, \ldots, g_k$. The coefficients are calculated via the inner product,

$$
c^\top = \begin{bmatrix} \langle f, g_1 \rangle \\ \vdots \\ \langle f, g_k \rangle \end{bmatrix} = \begin{bmatrix} \langle b^\top g, g_1 \rangle \\ \vdots \\ \langle b^\top g, g_k \rangle \end{bmatrix} = b^\top \begin{bmatrix} \langle g_1, g_1 \rangle & \cdots & \langle g_1, g_k \rangle \\ \vdots & \ddots & \vdots \\ \langle g_k, g_1 \rangle & \cdots & \langle g_k, g_k \rangle \end{bmatrix}
$$

The loss function $L = |f - \hat{f}|^2 = |f - c^\top g|^2$. If $c = b$, then the loss will be 0. Observe that $c = b$ if and only if the Gram matrix is identity, and the Gram matrix is identity only for an orthonormal basis. In other words, the minimizer of the loss function is an orthonormal basis. Thus, in order for gradient descent to decrease loss, the basis functions converge towards orthonormality [14]. This intuition is empirically validated in [14], Appendix A.5.

As a final note, the coefficients can be computed via least squares after training. Least squares does not require an orthonormal basis as it uses the Gram matrix to account for the inner products between basis functions.

