# OpenReview forum: "Zero-Shot Transfer of Neural ODEs"
_NeurIPS.cc/2024/Conference — NeurIPS 2024 poster_

### Official Review · Reviewer_w5ay · 2024-07-02

**Soundness:** 3
**Presentation:** 3
**Contribution:** 3
**Rating:** 6
**Confidence:** 4

**Summary:**

The work explores the use of neural ODEs as basis functions for function encoders. This requires dealing with the additional integration step, with the weighted combination of obtained ODEs representing the behaviour of the function to approximate via its integral. With an inner product and its tractable Monte Carlo estimation scheme, the authors derive an algorithm to train basis functions which span the space of dynamical systems, where each basis function is a neural ODE. The FE + ODE approach is put to the test against ODE and FE schemes for fitting a space of Van der Pol systems, long term prediction for RL, as well as quadrotor MPC control with improvements in zero-shot performance against the baselines used.

**Strengths:**

The authors deal with a very important problem in machine learning and all the more in robotics: that of zero-shot generalisation to changes in dynamics/environment (here modelled via hidden parameters)

- Interesting extension of the FE paradigm with a clear derivation of the required additions to adapt the basis function algorithm to neural ODEs
- Good readability and presentation
- Relatively significant combination of neural ODEs and Function Encoders with interesting performance improvements

**Weaknesses:**

- Some clumsiness in the math notation: in equation (3) $N$ is used at the denominator when it should be $m$ instead.
- The assumption of orthogonality for the validity of the coefficients is vital. However, both the authors in [13] and in this paper do not share any analysis on the validity of this assumption. It would be interesting to see how well this holds in practice (computing inner product of obtained basis functions/ODEs should be easy) and how many iterations are required to reach orthogonality in the examples provided.
- The introductory paragraph of section 3 where the link between a new unknown function $f$, its observed trajectory $\mathcal{D}$ and ODE basis functions is made, could benefit from more motivation and explanation. It is not immediately clear why reasoning at the derivative level while dealing with integral trajectory data is more advantageous. Perhaps a figure would help as the manuscript is text heavy.
- Explanations of the MuJoCo results are not very clear to me, they seem more like observations than interpretations. Why is the oracle unstable (does it require more data and training given the conditioning on hidden params)? What explains the differences between both experiments: FE + RES does well on half cheetah (better than FE + NODE) and terrible on ant? Where does the lack of inductive bias intervene in this case ?

**Questions:**

- The term zero-shot does indeed refer to the ability of networks to perform in novel circumstances without retraining, and does apply here. However, it can be slightly misleading as adaptability here requires analysis of new data from the new setting. Can the authors elucidate the scales involved in the tradeoff? In other words, when does it become interesting to train 100 base models (on the 100 datasets) to gain a deployment advantage that still requires new data to function, versus fine-tuning one model or retraining?
- Do the authors have any insights on changes in levels of performance whether the new ODE is in the convex hull of the available basis or outside it? (for example in the Van der Pol example, if the basis datasets contain trajectories for values of $\mu$ between 0.1 and 3, can the system perform for $\mu = 5$ ?

**Limitations:**

- A major numerical challenge working with continuous time neural networks is that of integration, with a plethora of solvers and schemes available. The authors touch upon this topic in their limitations sections and the impact of the integration horizon selection on behaviour predictability as well as the compute overhead involved (also in appendix C). It would have been interesting to give readers a better sense about the tradeoffs with numerical comparisons that go beyond verbal description.

---

> ### Author Rebuttal · Authors · 2024-08-02
>
> **The assumption of orthogonality for the validity of the coefficients is vital. However, both the authors in [13] and in this paper do not share any analysis on the validity of this assumption. It would be interesting to see how well this holds in practice (computing inner product of obtained basis functions/ODEs should be easy) and how many iterations are required to reach orthogonality in the examples provided.**
>
> Please see the response to all reviewers, section 3. Also section A.5 in [13] may be of interest.
>
> **The introductory paragraph of section 3 where the link between a new unknown function $f$, its observed trajectory $D$, and ODE basis functions is made, could benefit from more motivation and explanation. It is not immediately clear why reasoning at the derivative level while dealing with integral trajectory data is more advantageous. Perhaps a figure would help as the manuscript is text heavy.**
>
> To clarify, in continuous time dynamical systems, the dynamics is inherently a differential equation that is being integrated through time. However, it is typically the case that system observations occur at discrete time intervals, and do not include derivative measurements. Hence, neural ODEs are advantageous because they allow you to learn a model of the true system derivatives while using integral trajectory data.
>
> **Explanations of the MuJoCo results are not very clear to me, they seem more like observations than interpretations. Why is the oracle unstable (does it require more data and training given the conditioning on hidden params)? What explains the differences between both experiments: FE + RES does well on half cheetah (better than FE + NODE) and terrible on ant? Where does the lack of inductive bias intervene in this case ?**
>
> The apples-to-apples comparison is between FE + Res. and FE + NODE + Res. These two approaches differ only in basis function architecture, and FE + NODE + Res. consistency outperforms. The other relevant comparison is FE + NODE + Res. and the Oracle. These two approaches both benefit from the inductive bias of neural ODEs, and both have information on the hidden parameters. However, the format of the information is different. Oracle gets this information explicitly as an additional input, whereas FE + NODE +Res. implicitly gets this information via the coefficients of the basis functions. We expect that Oracle is performing badly because it is generalizing to an entire space of dynamics with one NODE, which is a very complex function to learn. We believe that Ant is much more difficult and varied than Cheetah, and the relative ease of Cheetah is likely why both Res. methods perform so well on Cheetah. We have improved the discussion in section 4.2 to include these details.
>
> **The term zero-shot does indeed refer to the ability of networks to perform in novel circumstances without retraining, and does apply here. However, it can be slightly misleading as adaptability here requires analysis of new data from the new setting. Can the authors elucidate the scales involved in the tradeoff? In other words, when does it become interesting to train 100 base models (on the 100 datasets) to gain a deployment advantage that still requires new data to function, versus fine-tuning one model or retraining?**
>
> Two things to note. Rather than training a function for every dataset, we train $k$ basis functions to span all datasets. So, if we have 100 datasets but only want to train 10 basis functions, this is possible without issue. Secondly, it does require a small amount of online data. This data is necessary for any method because you need some information to identify the current dynamics. For example, finetuning a neural ODE  based on online data is another approach, but this also requires online data. Indeed, finetuning a model likely requires even more data than our approach, and incurs a significant computational overhead that is going to take significantly longer than our approach. In contrast, we can compute the coefficients of the basis functions in milliseconds, as the inner product calculation is effectively a sample mean, and then we can instantly adapt our predictions based on the data.
>
> **Do the authors have any insights on changes in levels of performance whether the new ODE is in the convex hull of the available basis or outside it? (for example in the Van der Pol example, if the basis datasets contain trajectories for values of $\\mu$ between 0.1 and 3, can the system perform for $\\mu=5$?**
>
> Examining OOD generalization is an interesting future direction. Please see the response to all reviewers, section 2, for the experiment you described.
>
> **A major numerical challenge working with continuous time neural networks is that of integration, with a plethora of solvers and schemes available. The authors touch upon this topic in their limitations sections and the impact of the integration horizon selection on behaviour predictability as well as the compute overhead involved (also in appendix C). It would have been interesting to give readers a better sense about the tradeoffs with numerical comparisons that go beyond verbal description.**
>
> We leverage RK4, and there are indeed more accurate integrators available. However, there is inherently a trade off with respect to both training time and execution time. Integrators such as adaptive step size solvers can potentially make 20 or more calls to the neural ODE during a forward pass, while RK4 makes only 4. The increased number of neural ODE forward passes greatly increases memory usage and compute time. We experimented with better integrators, but ultimately found this tradeoff to be unfavorable. We find RK4 to be the right balance of speed and accuracy for this work. We have added an additional section of the appendix discussing the choice of integrator.

---

> > ### Comment · Reviewer_w5ay · 2024-08-12
> >
> > I thank the authors for the elements presented in the rebuttal and general response. They clear up some points of confusion and improve in my opinion the understanding of the work, although they do not shed light on new strengths of the approach that might have gone unnoticed in my initial review. Thus, the current score still represents my appreciation of the work.

---

### Official Review · Reviewer_6dAJ · 2024-07-08

**Soundness:** 3
**Presentation:** 3
**Contribution:** 3
**Rating:** 5
**Confidence:** 4

**Summary:**

The paper presents a novel framework for the zero-shot transfer of neural ODEs by leveraging function encoders to represent a space of dynamical systems. It demonstrates the method's effectiveness in adapting to unseen environments without retraining, using MuJoCo and quadrotor experiments.

**Strengths:**

The paper demonstrates interesting ideas by introducing a novel framework for zero-shot transfer of neural ODEs using function encoders. This approach enables adaptation to unseen scenarios without retraining using the neural ODEs and function encoders. This research shows its potential to enhance the adaptability and safety of autonomous systems, bridging the gap between training and testing data.

**Weaknesses:**

The paper presents a promising framework for zero-shot transfer of neural ODEs, but there are several areas for improvement. Firstly, the reliance on a large and diverse dataset for training is a significant limitation. The approach requires extensive data that spans the entire function space of possible dynamics, which may not always be feasible. This dependency on comprehensive datasets should be addressed by exploring data-efficient learning methods or leveraging transfer learning techniques.

Secondly, the computational overhead involved in training multiple neural ODEs is substantial. This might hinder the scalability and real-time applicability of the proposed method. The authors could investigate more efficient training algorithms or consider approximations that reduce computational costs without compromising accuracy.

Thirdly, the paper does not explicitly enforce the orthogonality of basis functions, relying instead on implicit regularization through the loss function. This might lead to suboptimal representation of the function space, affecting the model's performance.

Moreover, the experiments lack diversity in evaluating the model's robustness to completely unseen environments or significantly different conditions. Including more varied test scenarios or stress tests could provide a deeper understanding of the model's adaptability and limitations.

**Questions:**

1. The computational overhead for training multiple neural ODEs might be significant. How does this affect the scalability and real-time applicability of your method?

2. The paper does not enforce orthogonality explicitly for the basis functions, which might lead to suboptimal representation. Have you evaluated the levels of orthogonality (and their relationship with performance) in your method?

3. The experiments lack diversity in evaluating the model's robustness to completely unseen environments or significantly different conditions. How confident are you in your model’s adaptability in such scenarios?

4. The evaluation metrics used in the MuJoCo experiments are primarily focused on prediction accuracy. Have you considered additional metrics that might better capture your model's practical performance?  For example,  task-specific performance such as scores or cumulative rewards.

5. It seems to lack a sufficient variety of baselines for comparison. Have you considered additional zero-shot / few-shot or multitask / meta-learning studies?

**Limitations:**

Refer to Weaknesses and Questions.

---

> ### Author Rebuttal · Authors · 2024-08-02
>
> **The paper presents a promising framework for zero-shot transfer of neural ODEs, but there are several areas for improvement. Firstly, the reliance on a large and diverse dataset for training is a significant limitation. The approach requires extensive data that spans the entire function space of possible dynamics, which may not always be feasible. This dependency on comprehensive datasets should be addressed by exploring data-efficient learning methods or leveraging transfer learning techniques.**
>
> Any approach that aims to model a large set of dynamics from online data inherently must learn from a large dataset. This is because the space of functions is infinite dimensional, and this large, diverse dataset is effectively teaching the model which subspace is most important. Therefore, the ability to infer behavior from data necessitates the presence of a large training dataset.
>
> Our approach learns basis functions to span the functions present in the training set. However, it’s important to note that any function in the span of the basis can be perfectly represented, and thus our approach is an efficient transfer learning technique for this domain. However, examining strategies for further improving data efficiency is an interesting future direction. For example, it may be possible to augment the training dataset with additional common functions, such as linear transformations between inputs and outputs or trigonometric functions, as this would increase the diversity of the training dataset without requiring more data.
>
> **Secondly, the computational overhead involved in training multiple neural ODEs is substantial. This might hinder the scalability and real-time applicability of the proposed method. The authors could investigate more efficient training algorithms or consider approximations that reduce computational costs without compromising accuracy.**
>
> Overhead is indeed an important consideration, and we have two strategies to address this. During the offline training phase, the key consideration is memory usage. This is because neural networks converge faster with larger batch sizes, and so leveraging the largest batch size possible is key. To address this, we run the basis functions sequentially during training. Since only one basis function is being called at a time (both during the forward pass and back-propagation), the memory overhead is the same as a single neural ODE. This does increase the training time by a factor of $k$. However, as neural ODEs alone cannot model multiple dynamics functions, we view this cost as acceptable for the benefits our approach provides.
>
> During the online phase, memory usage is inherently reduced as all memory usage results from online data only. So, memory is not a concern. Instead, the compute speed is the most important factor for real-time control. To address this, we run the basis functions in parallel. As there is no dependence between them, each basis function can run at the same time. Thus, by running the basis functions in parallel, there is no additional compute time overhead. Therefore, this method is equivalent to a vanilla neural ODE with respect to real-time compute speeds.
>
> We have improved the discussion in the appendix, section C, to include this additional information.
>
> **Thirdly, the paper does not explicitly enforce the orthogonality of basis functions, relying instead on implicit regularization through the loss function. This might lead to suboptimal representation of the function space, affecting the model's performance.**
>
> Please see the Response to all Reviewers, section 3.
>
> **Moreover, the experiments lack diversity in evaluating the model's robustness to completely unseen environments or significantly different conditions. Including more varied test scenarios or stress tests could provide a deeper understanding of the model's adaptability and limitations.**
>
> Verifying OOD generalization capabilities is an interesting future direction. Please see the response to all reviewers, section 2.
>
> **The experiments lack diversity in evaluating the model's robustness to completely unseen environments or significantly different conditions. How confident are you in your model’s adaptability in such scenarios?**
>
> Please see the response to all reviewers, section 2. It’s worth noting that the test environments are sampled from the same distribution of environments, but are unseen by the model during training.
>
> **The evaluation metrics used in the MuJoCo experiments are primarily focused on prediction accuracy. Have you considered additional metrics that might better capture your model's practical performance? For example, task-specific performance such as scores or cumulative rewards.**
>
> The MuJoCo experiments are designed specifically to highlight prediction accuracy. We agree that other metrics might be useful as well, and so we leverage a task-specific efficiency metric in the quadrotor experiments. For more information see section 4.3, especially the last paragraph.
>
> **It seems to lack a sufficient variety of baselines for comparison. Have you considered additional zero-shot / few-shot or multitask / meta-learning studies?**
>
> We are unaware of any other zero-shot techniques for neural ODEs. It is possible to use a transformer for the same type of data, however transformers suffer notable drawbacks. First, their memory usage is quadratic, and so they are unable to make use of large, online datasets. Second, their forward pass time is quite slow, and so they are not suitable for real time model-predictive control as many forward passes must be made every timestep, e.g. 100s of forward passes in 30 milliseconds. Meta-learning techniques are similar in that they adapt models given instance-specific data. However, these techniques perform additional finetuning or retraining based on this data, which makes them ill-suited for real-time control.

---

> ### Comment · Reviewer_6dAJ · 2024-08-13
>
> The authors' rebuttal and general response have improved my understanding and addressed many of my concerns. As a result, I have revised my score to 5.

---

### Official Review · Reviewer_3pa5 · 2024-07-12

**Soundness:** 3
**Presentation:** 4
**Contribution:** 2
**Rating:** 5
**Confidence:** 4

**Summary:**

This paper proposes a method to learn the dynamics of autonomous systems in a few shot manner. The core assumption is that the dynamics function dx/dt=f(x) of a new system can be modeled by a linear combination of basis dynamics functions. The method involves two stages. In the offline stage, the method learns a set of basis functions with neural ODE. In the online stage, the method uses incoming dynamics to fit the linear weights, which when combined with the learned basis, can be used to model the new system. The authors evaluated the effectiveness of the method on  Van der Pol Oscillato system (with varying parameters) and Mujoco Ant.

**Strengths:**

1. The writing is very clear. Readers with reasonable math background & ODE shall understand this paper reasonably well

2. The core of the method is intuitive, and the method is sound. Basis function with linear weighting for quick adaptation has been grounded in many fields.

3. I appreciate the fact that the authors explains the limitation and scope of the project, and there is no overclaim

**Weaknesses:**

1. My main criticism of the paper is the technical contribution. The problems this system can solve seem to be constrained to systems limited variation in parameter, where the offline dataset & online system share a high level of similarity. While the authors explained how they apply NODE very clearly, I think using NODE is not a fundamental contribution because you can also have other sequence models trained to predict residue. The idea of basis function has also been explored by many prior literatures.

2. The title is misleading - you are doing quick online adaptation (few shot) instead of a zero-shot setting. A few trajectories of online data is required to adapt to the new system

3. There are a lot of baselines the authors should look into since few shot learning / quick adaptation is a long standing topic of research. For example, other basis function methods, meta learning methods. I understand that the authors picked a setting where time is continuous, but the mujoco Ant environment is also traditionally studied as discrete too, so a meta learning + some sequence prediction model should be applicable here. If you can provide a convincing argument about why those priors works aren't applicable it's also fine.

4. As I suggested in questions section, there could be many perspectives the author should dive deeper to improve the paper


minor:
1. In line 114, the reasoning doesn't provide grounding to the orthogonality question - the training objective make them span the space assuming diverse enough dynamic systems - but this is unrelated to orthogonality! Are you saying that non orthogonal is okay as soon as they span the space?

2. It would be good to give more intuitive visualizations of estimation error in mujoco Ant. e.g. render the predicted trajectories as video

3. ablation about the number basis functions used would be helpful to understanding

**Questions:**

1. In line 165, the authors said "Given data collected online from a single trajectory". I am curious to see ablations how the method improves as the amount of online data grow

2. In figure 3, error goes up as the number of look-ahead steps increases. While this is partially expected due to compounding error, how much of it shall be attributed to limited expressiveness of linear basis. e.g. I can do an experiment where I train a NODE with many many data, not just 200 example points, and roll it out. You will witness how MSE loss increases and gain insights about a upper bound for prediction accuracy that's not due to limited data / limited number of basis

**Limitations:**

Please see my points in weakness

---

> ### Author Rebuttal · Authors · 2024-08-02
>
> **My main criticism of the paper is the technical contribution. The problems this system can solve seem to be constrained to systems limited variation in parameter, where the offline dataset & online system share a high level of similarity. While the authors explained how they apply NODE very clearly, I think using NODE is not a fundamental contribution because you can also have other sequence models trained to predict residue. The idea of basis function has also been explored by many prior literatures.**
>
> * We use hidden environmental parameters as an example problem setting because it is frequent in robotics. However, we never make use of the hidden parameters directly. As mentioned in Section 3, this algorithm can be applied to a space of dynamical systems that arises for *any* reason.
>
> * Generalization to OOD online systems is indeed an interesting direction. Please see section 2 of the response to all reviewers for a discussion.
>
> * We are interested in model-based control. Sequence prediction models, e.g. transformers, are often too computationally complex for model predictive control. SOTA control algorithms such as iCEM MPC need hundreds of forward passes per control-action. Transformers are well known to have quadratic cost with respect to input space, and their forward pass is much slower than neural ODEs, and so they are not suitable for this setting. Meta-learning algorithms can be applied under the same data assumptions, but often finetune or retrain the learned model based on new data. This training period is not amenable to real-time control. Lastly, there is indeed a plethora of analytical basis function methods. However, analytical basis functions often scale extremely poorly to high-dimensional function spaces. Other approaches like SINDy have similar scaling issues, but also require a potentially lengthy sparse coefficient identification procedure which can be too time-consuming for real time control. In contrast, our method enjoys the same mathematical interpretation as these approaches, while scaling to high-dimensional problems due to the use of a relatively small number of neural network basis functions. We have extended the related works to include information on why common sequence models and meta-learning are not applicable to this setting.
>
> **The title is misleading - you are doing quick online adaptation (few shot) instead of a zero-shot setting.**
>
> The term “zero-shot” means different things to different communities. For example, the LLM community will often say “few-shot prompting” to mean operating a model with a few examples as input, and no finetuning. However, for ODEs, dynamics modeling, and reinforcement learning, “Zero-shot” is the ability to leverage unseen data at execution time to improve model performance without additional training, where the zero comes from zero gradient steps. See “Does Zero-Shot Reinforcement Learning Exist?” (Ahmed Touati, Jérémy Rapin, Yann Ollivier, 2022), “Unsupervised Zero-Shot Reinforcement Learning via Functional Reward Encodings“ (Kevin Frans, Seohong Park, Pieter Abbeel, Sergey Levine, 2024), “Zero-Shot Reinforcement Learning via Function Encoders” (Tyler Ingebrand, Amy Zhang, Ufuk Topcu, 2024), etc.
>
> **A few trajectories of online data is required to adapt to the new system.**
>
> For the Mujoco examples, we only leverage 200 data points. For a system running at 30 Hz, this is equivalent to only ~7 seconds of data rather than a full trajectory. This data can then be used to identify the system dynamics immediately at runtime.  We have improved the experiment description in section 4.2 to clarify this.
>
> **In line 114, the reasoning doesn't provide grounding to the orthogonality question, ...,  Are you saying that non orthogonal is okay as soon as they span the space?**
>
> Thank you for your comment, we have revised the wording of line 114 for clarity. Please see the response to all reviewers, section 3 for more information on orthonormality.
>
> **It would be good to give more intuitive visualizations of estimation error in mujoco Ant. e.g. render the predicted trajectories as video.**
>
> Good idea, we will add screenshots to the appendix.
>
> **Ablation about the number basis functions used would be helpful to understanding.**
>
> We agree that the effect of the number of basis functions is an interesting ablation. Please see the response to all reviewers, section 1a.
>
> **In line 165, the authors said "Given data collected online from a single trajectory". I am curious to see ablations how the method improves as the amount of online data grow.**
>
> We agree that examining the prediction accuracy as the amount of data increases is an interesting experiment. Please see the response to all reviewers, section 1b.
>
> **In figure 3, error goes up as the number of look-ahead steps increases. While this is partially expected due to compounding error, how much of it shall be attributed to limited expressiveness of linear basis. e.g. I can do an experiment where I train a NODE with many many data, not just 200 example points, and roll it out. You will witness how MSE loss increases and gain insights about a upper bound for prediction accuracy that's not due to limited data / limited number of basis.**
>
> Please see the response to all reviewers, section 1, which may provide some more intuition on the effect of the example data and number of basis functions on the performance. While using more example data slightly improves performance, a key feature of our approach is that it can operate in low data settings. Additionally, the method is insensitive to the number of basis functions, which implies that the basis functions are more than sufficient to express the dynamics. Lastly, “NODE” and “Oracle” are also neural ODE models, and so they suffer from the same compounding error problems as our approach. Thus, those approaches provide a reference for how much of the error is due to the inherent difficulty of dynamics prediction.

---

> > ### Comment · Reviewer_3pa5 · 2024-08-09
> >
> > I acknowledge that I've read the rebuttal and general response. Thank you for the ablations and they are helpful to understanding. Here are some additional questions:
> >
> > 1. If quality of prediction isn't sensitive to number of basis functions & data after they are increased to a certain degree, does it mean that the method won't be able to scale up further to bring the prediction error to 0, even for a deterministic system. I understand that every predictive model have errors, I am simply looking for some analysis here - what's the source of the remaining error.
> >
> > 2. In my review, I mentioned that
> >
> > > The problems this system can solve seem to be constrained to system limited variation in parameter
> >
> > While the authors did an ablation on OOD parameters, it's still limited to the family of OOD that's reflected via parameters. I am wondering whether the system can generalize to OOD dynamics that completely changes its form (e.g. walking to jumping), not just parameters

---

> > > ### Author Response · Authors · 2024-08-10
> > >
> > > **If quality of prediction isn't sensitive to number of basis functions & data after they are increased to a certain degree, does it mean that the method won't be able to scale up further to bring the prediction error to 0, even for a deterministic system. I understand that every predictive model have errors, I am simply looking for some analysis here - what's the source of the remaining error.**
> > >
> > > This is a nuanced and complicated question. The biggest thing that likely leads to error is the fact that the system (MuJoCo) is not strictly continuous due to contact forces. Therefore, there will always be some error for any method that uses a neural network due to the contact forces in the environment. Another factor is that the space of functions is infinite-dimensional. Therefore, for any finite number of basis functions, there is inherently going to be error due to the unrepresented dimensions. We do expect to see diminishing returns as the number of basis functions increase, as some of these dimensions are less important for predicting the dynamics than others, and this aligns with the empirical results. We also leverage RK4 as the integrator for the neural ODE. RK4 is a fourth-order method, and higher-order terms are truncated. This choice of integrator therefore imposes error, which increases for larger time horizons. More accurate integrators are available, but come at the cost of compute time. Lastly, numerical precision may play a small role here due to implementation details. Any numerical precision errors will compound as more operations occur. These errors can affect both training, where they effectively appear as noise added to the gradients, and during execution.
> > >
> > > **In my review, I mentioned that “The problems this system can solve seem to be constrained to system limited variation in parameter”. While the authors did an ablation on OOD parameters, it's still limited to the family of OOD that's reflected via parameters. I am wondering whether the system can generalize to OOD dynamics that completely changes its form (e.g. walking to jumping), not just parameters**
> > >
> > > No, our approach is not limited to variation in parameters. As stated in the paper in section 3, we are able to handle variations due to changes in the underlying physics model. The only theoretical requirement is that the functions exist in the same space. We believe the MuJoCo examples demonstrate this as our model can generalize beyond simple parameter varying systems, where the robot’s physical shape is changing. However, your question highlights two nuanced questions. The first is a shift in the distribution of states seen between a robot that is walking and jumping. A jumping robot is likely to experience different states than when it is walking. If these states have never been trained on, then any learned model cannot hope to accurately model these dynamics (without leveraging prior knowledge). In other words, a purely learned model which leverages no prior information needs its training set to cover the state space. The second question is if the learned dynamics can generalize across behaviors. Our model has actions as an input, and therefore it can accurately predict a given transition even if the underlying policy was not used to collect training data. Therefore, generalization across behaviors is possible.

---

> > > > ### Comment · Reviewer_3pa5 · 2024-08-10
> > > >
> > > > The authors mentioned that transformers are often too computationally complex for model predictive control, but doesn't the same apply to Neural ODE too? From my personal experience, NODE isn't quite fast either, and depending on the landscape of the dx/dt, solver can struggle quite a bit, especially if the authors model a second order system e.g. mujoco locomotion environments' observations are often positions + velocities. Have you tried transformer style architecture for the problems?

---

> > > > > ### Author Response · Authors · 2024-08-10
> > > > >
> > > > > Many prior works have addressed this question, and leverage neural ODEs for model-predictive control. For example, please see “Taylor-lagrange neural ordinary differential equations: Toward fast training and evaluation of neural odes” (Djeumou, Neary, Goubault, Putot, Topcu, 2022) which proposes a fixed step size method to that achieves a forward pass in 1 millisecond or less. See also “How to Learn and Generalize From Three Minutes of Data: Physics-Constrained and Uncertainty-Aware Neural Stochastic Differential Equations” (Djeumou, Neary, Ufuk Topcu, 2023), “Autonomous Drifting with 3 Minutes of Data via Learned Tire Models” (Djeumou, Goh, Topcu, Balachandran, 2023),  “Learning-enhanced Nonlinear Model Predictive Control using Knowledge-based Neural Ordinary Differential Equations and Deep Ensembles” (Chee, Hsieh, Matni, 2023), “Model predictive control of nonlinear processes using neural ordinary differential equation models” (Luo, Abdullah, Christofides 2023).
> > > > > For adaptive step size solvers, the number of forward passes required to integrate a given time horizon depends on the landscape of dx/dt, as you mention. Indeed, this is a serious limitation of adaptive step size solvers, as the solver can require 10-100 forward passes per integration. However, other fixed step size solvers exist, such as higher-order RK methods or Taylor-Lagrange methods, and these methods may achieve better accuracy without incurring significant overhead. In our work, we use RK4, which has only 4 forward passes per integration step, meaning the compute time is much quicker, and we find that this tradeoff is favorable for real-time control.
> > > > >
> > > > > Transformer-based MPC is a nascent topic, and has been explored in “Simultaneous multistep transformer architecture for model predictive control”, (Park et. al., 2023), which shows that transformers achieve significant speedups vs. LSTM-based architectures, but are still longer than required for robotics applications (on the order of seconds). In our own observations, transformer-based architectures are too computationally inefficient since they must compute attention on the example data for every forward pass, in a computation that scales quadratically with the amount of data.
> > > > >
> > > > > We have improved the discussions in Section 3.2 and in the Appendix to highlight these details.

---

> > > > > > ### Comment · Reviewer_3pa5 · 2024-08-12
> > > > > >
> > > > > > Overall the author did a good job during the rebuttal. The added ablations addressed most of my concerns. I believe that giving readers more insights via ablation is one direction the paper could further improve. I am changing my score to "technical solid paper, accept outweigh reasons to reject", but I'd still expect more impact from a paper for any higher scores.
> > > > > >
> > > > > > I strongly encourage the authors to revise their paper to include a detailed discussion about orthonormality, number of basis functions & data in main paper, and probably NeuralODE speed in the appendix..

---

> > > > > > > ### Author Response · Authors · 2024-08-12
> > > > > > >
> > > > > > > We thank the reviewer for their insightful comments and discussion, which has improved our central arguments and clarified several details. We respectfully maintain that the ability to adapt and generalize in real time, given a small online dataset, has significant impact for the field of autonomy and robotics. Online adaptability is clearly necessary to bring learning-based systems from laboratory settings to the unstructured nature of real world environments. To the best of our knowledge, we provide the first scalable algorithm for learning models of dynamical systems that can adapt near-instantaneously at runtime without any gradient updates and without leveraging prior information. We believe this will have significant impact on the field of autonomy and robotics.

---

### Official Review · Reviewer_A5qk · 2024-07-13

**Soundness:** 3
**Presentation:** 3
**Contribution:** 3
**Rating:** 6
**Confidence:** 2

**Summary:**

The paper aims to address the challenge of zero-shot transfer and adaptation. The authors propose tackling this challenge by learning a dynamics space spanned by neural ODE basis functions, which can then be used for rapid identification and adaptation to dynamics at inference time without additional training. The paper demonstrates the efficacy of the proposed approach, using both a simpler oscillator system and scaled up simulated robotics environments.

**Strengths:**

- While this is not my area of expertise, to the best of my knowledge, the method proposed in the paper and the corresponding experiments are novel.
- The paper is well written and clear, and fast adaptation is a crucial problem to the field, especially in the field of robotics and control.
- The experiments seem well designed and thorough, showing the contribution of each of the method components when ablated, and demonstrating the efficacy of the approach for control via MPC.

**Weaknesses:**

- To place the results in context with prior work, it would be additionally helpful for the robotics experiments to show comparisons to other methods that enable adaptation (e.g. training a model free method with domain randomized parameters).
- It’s not clear how well the method will scale to more complex dynamics, or when trying to generalize beyond the learned basis functions (e.g., if they are not expressive enough or span the relevant spaces for a dynamical system, or if insufficient data is used to learn the dynamics space).

**Questions:**

See suggestions in the Weaknesses section above.

**Limitations:**

The authors address method limitations adequately in the paper.

---

> ### Author Rebuttal · Authors · 2024-08-02
>
> Thank you for your feedback. See below for responses to your suggestions.
>
> **To place the results in context with prior work, it would be additionally helpful for the robotics experiments to show comparisons to other methods that enable adaptation (e.g. training a model free method with domain randomized parameters).**
>
> Domain randomization allows RL algorithms to find policies which are robust to randomized parameters, however the policy does not adapt to those parameters. Our approach is inherently solving a different problem, in that our controller can adapt to the specific dynamics it is experiencing. This is done by using a small, online dataset to identify the coefficients of the current dynamics function. Then, the basis functions (in combination with the coefficients) are used to model the current system dynamics. A controller using this system model therefore adapts to the current dynamics in an online fashion.
>
> **It’s not clear how well the method will scale to more complex dynamics, or when trying to generalize beyond the learned basis functions (e.g., if they are not expressive enough or span the relevant spaces for a dynamical system, or if insufficient data is used to learn the dynamics space).**
>
> The ablation in the response to all reviewers, section 1a, shows that the basis functions are sufficiently expressive to span the space of dynamics present in the system. Furthermore, as $L_2$ is a Hilbert space, any function in  $L_2$ can be perfectly described as a linear combination of basis functions. In other words, basis functions are sufficiently expressive for the vast majority of problem settings.
>
> Generalization beyond the function space in the dataset is a fascinating question. Please see the response to all reviewers, section 2 for more information.

---

### Author Rebuttal · Authors · 2024-08-02

# Response to all Reviewers:

We thank the reviewers for their comments and keen insights. We have made the following major changes to the paper in response to their feedback.

## **1. Hyper-Parameters**:

We have added an ablation on both the number of basis functions and the number of example data points.

### **1a. Number of Basis Functions**:

We ablate different numbers of basis functions at predicting dynamics in the Half-Cheetah environment. We find that the algorithm is generally insensitive to the number of basis functions, and performance eventually decays as the number of basis functions approaches 0. See the attached PDF for the corresponding figure. The results indicate that 100 basis functions are sufficiently expressive for the dynamics of the ablated environment.

### **1b. Number of Example Data Points**:

We ablate the sensitivity to the number of data points on predicting dynamics in the Half-Cheetah environment. We find that increasing data only slightly improves performance and that there are diminishing returns. See the attached PDF for the corresponding figure. Note that if the function is within the span of the basis, then its prediction accuracy only depends on the error between the true basis function coefficients and the Monte Carlo estimate of the coefficients. As the estimate is computed via a sample mean, the error in this prediction decreases to 0 as more data is collected due to the law of large numbers.


### **2. Generalization**:

For dynamics functions outside of the training set, it can be shown that any function within the span of the basis can be perfectly computed. Functions that lie outside the span of the basis may still have low error if they are sufficiently close to the learned subspace. To illustrate this point, we train the Van Der Pol example on $\mu \in [0.1, 3.0]$, and evaluate it on $\mu=4.0$.  We observe that the model is still able to reasonably approximate this out-of-distribution $\mu$. See the figure in the attached PDF. Exploring the exact OOD generalization capabilities of this algorithm, along with possible error bounds, is an active direction of future work, but is outside the scope of this paper.

### **3. Orthonormality**:

This discussion relates to the function encoder algorithm of the paper “Zero-Shot Reinforcement Learning via Function Encoders”. Nonetheless, we present the following discussion for clarity as it is relevant to this work.

Consider a set of basis functions $g_1, …, g_k$. Suppose that $g_1, …, g_k$ is not orthonormal. Consider a function $f$, and suppose $f$ happens to be in the span of $g_1, …, g_k$. Then $f$ can be expressed as $f=b^\top g$, where $b$ is a set of coefficients and $g$ is the concatenation of $g_1, …, g_k$. Consider the coefficients calculated via the inner product,

$c^\\top =\\begin{bmatrix} \\langle f, g_1 \\rangle \\\\ \\vdots \\\\ \\langle f, g_k \\rangle \\end{bmatrix} =\\begin{bmatrix} \\langle b^\\top g, g_1 \\rangle \\\\ \\vdots \\\\ \\langle  b^\\top g, g_k \\rangle \\end{bmatrix} =b^\\top \\begin{bmatrix} \\langle g_1, g_1 \\rangle & … &  \\langle g_1, g_k  \\rangle \\\\ \\vdots & \\ddots & \\vdots \\\\ \\langle g_k, g_1 \\rangle & … & \\langle  g_k, g_k \\rangle \\end{bmatrix}$

Consider the loss function $l=\\vert f - \\hat{f} \\vert ^2=\\vert f - c^\\top g \\vert ^2$. If $c=b$, then the loss will be 0. Observe that $c=b$ if and only if the Gram matrix is identity, and the Gram matrix is identity only for an orthonormal basis. In other words, the minimizer of the loss function is an orthonormal basis. Thus, in order for gradient descent to decrease loss, the basis functions converge towards orthonormality.
This intuition is empirically validated in “Zero-Shot Reinforcement Learning Via Function Encoders”, Appendix section A.5. Please see that paper for more information.

As a final note, as mentioned in section 2.2, line 106, the coefficients can be computed via least squares after training. Least squares does not require an orthonormal basis as it uses the Gram matrix to account for the inner products between basis functions.

In summary, it has been shown in prior works that the basis functions converge towards orthonormality. As stated in section 2.2, we may also use least squares at execution time, which sidesteps the issue. We have added this discussion to the appendix.

---

### Decision · Program_Chairs · 2024-09-25

**Decision:**

Accept (poster)

**Comment:**

This paper presents a neural ODEs-based method for modeling differential equations by learning basis functions, focusing on capturing system dynamics of autonomous systems within a structured function space.

The paper was reviewed by four reviewers who all are inclined to suggest for acceptance. The most reviewers agree that (1) the proposed method, the use of neural ODEs as function encoders, is intuitive and sound, (2) dealing with important problems, zero-shot generalization to dynamic environments, in machine learning and autonomous systems. The reviewers also commented that (3) the paper is well written with good readability and presentation.

Considering these strengths, this paper is recommended for acceptance. The authors are encouraged to utilize their constructive and detailed discussion with reviewers towards further raising the quality and impact of their work.